# MACM: Utilizing a Multi-Agent System for Condition Mining in Solving Complex Mathematical Problems

**Bin Lei**[1,2]  **Yi Zhang**[2]  **Shan Zuo**[2]  **Ali Payani**[3]  **Caiwen Ding**[1]

[1]University of Minnesota  [2]University of Connecticut  [3]CiscoResearch

{lei00126, dingc}@umn.edu
{bin.lei, yi.2.zhang, shan.zuo}@uconn.edu
{apayani}@cisco.com

## Abstract

Recent advancements in large language models, such as GPT-4, have demonstrated remarkable capabilities in processing standard queries. Despite these advancements, their performance substantially declines in **advanced mathematical problems requiring complex, multi-step logical reasoning**. To enhance their inferential capabilities, current research has delved into *prompting engineering*, exemplified by methodologies such as the Tree of Thought and Graph of Thought. Nonetheless, these existing approaches encounter two significant limitations. Firstly, their effectiveness in tackling complex mathematical problems is somewhat constrained. Secondly, the necessity to design distinct prompts for individual problems hampers their generalizability. In response to these limitations, this paper introduces the *Multi-Agent System for conditional Mining* (**MACM**) prompting method. It not only resolves intricate mathematical problems but also demonstrates strong generalization capabilities across various mathematical contexts. With the assistance of MACM, the accuracy of GPT-4 Turbo on the most challenging level five mathematical problems in the MATH dataset increase from $54.68\%$ to $76.73\%$. The code is available in https://github.com/bin123apple/MACM.

## 1   Introduction

Large language models (LLMs) like GPT-4 [12] excel in various problem-solving tasks but still struggle with complex logical deduction, especially in mathematics involving abstract concepts and multi-step reasoning [11, 2]. This limitation hinders their accuracy and reliability in fields requiring precise mathematical reasoning, such as academic research, engineering, and theoretical physics.

A contemporary and efficacious method for tackling this issue is the *prompting engineering* [19]. It enhances accuracy in complex problem-solving without necessitating further training of the model. By strategically crafting prompts, prompting engineering optimizes the utilization of large language models, guiding their processing pathways more efficiently and effectively [10].

Previous prompting methods mainly include the Chain of Thought (CoT) [18], Self-consistency Chain of Thought (SC-CoT) [17], Tree of Thought (ToT) [20], and Graph of Thought (GoT) [4, 9]. CoT and SC-CoT show limited capabilities in complex logical reasoning, achieving only 4.0% and 9.0% accuracy in simple tasks like the 24-point game using GPT-4 [20]. While ToT and GoT have improved LLMs' problem-solving abilities, they lack generalizability, requiring specific prompts for each problem, as detailed in Appendix A.

**To address two key issues:**

1. *The insufficient reasoning capability of LLMs for complex mathematical problems.*
2. *The inadequate generalizability of current prompting methods.*

38th Conference on Neural Information Processing Systems (NeurIPS 2024).

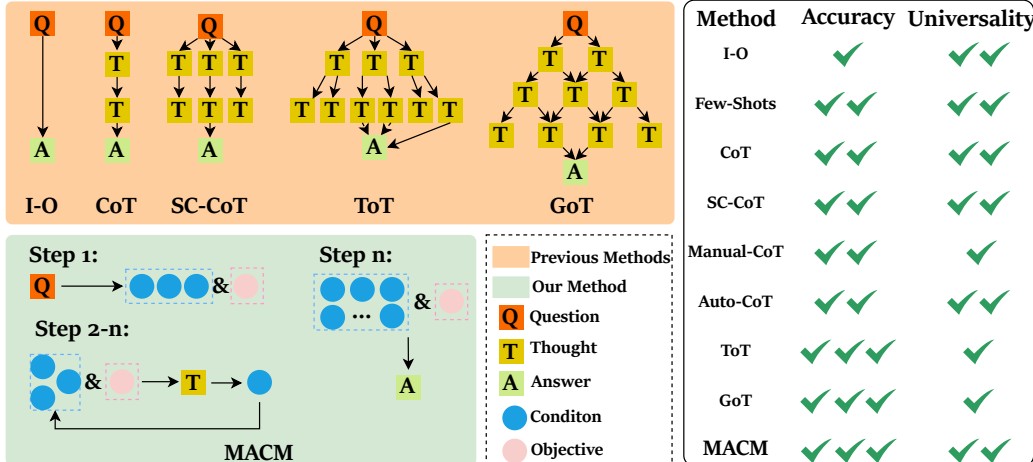

Figure 1: The comparison between the current mainstream prompting methods and MACM.

We propose the *Multi-Agent System for Condition Mining (MACM)* prompting method. MACM has moved beyond being restricted by the specific contents of a problem. Instead, it first abstracts the conditions and the objective of the problem. Subsequently, through a Multi-Agent interactive system, it progressively mines new conditions conducive to achieving the objective, thereby ultimately fulfilling the goal of problem-solving.

The comparison between MACM and current mainstream prompting methods in problem-solving is shown in Figure 1. MACM extracts conditions and objectives from each math problem, iteratively adding new insights until enough information is gathered to find a solution. Performance-wise, MACM improves accuracy by over 10 percentage points on datasets like the 24-point game, matching the effectiveness of ToT and GoT. Moreover, MACM is versatile; it can apply the same set of prompts to various mathematical reasoning problems without manual modifications, unlike the tailored prompts needed for ToT and GoT.

Through several experiments on the math related datasets, we verified MACM's generalizability and superior error correction compared to original prompting methods. With MACM, the GPT-4 turbo model's accuracy on the MATH dataset increased by $15.14\%$, and by $7.8\%$ compared to SC-CoT. In the 24-point game, MACM achieved an accuracy rate $17\%$ higher than ToT.

## 2 Related Work

In this section, we summarize several major current prompting methods.

***I-O* Prompting:** Input-Output (I-O) prompting is the most common method for interacting with large language models, where users specify the problem conditions directly to the model, which generates responses through a token-level, sequential decision-making process [22].

***CoT* Prompting:** [18]: Chain of Thought (CoT) prompting refines the model's output into more structured and logically coherent text by methodically constructing and elaborating upon chains of reasoning. This approach enhances the model's ability to produce outputs rooted in logical deduction. There are several variants of CoT, including Zero-Shot CoT [8], Few-Shot CoT [7], and Auto-CoT [22], each tailored to different prompting scenarios and requirements to improve logical reasoning in diverse contexts.

***SC-CoT* Prompting:** [17]: Self-Consistency Chain of Thought (SC-CoT) prompting improves upon CoT by introducing a voting mechanism, which emphasizes internal consistency and semantic interconnectedness. In this method, models evaluate (vote on) their own outputs to select the most coherent response, thereby reducing logical fallacies and inconsistencies.

***ToT* Prompting:** [20]: Tree of Thought (ToT) prompting uses a hierarchical, tree-like structure to organize and guide the model's text generation. This method improves precision and structure in responses, incorporating a voting mechanism to refine outcomes and reduce computational demands.

***GoT* Prompting :** [4, 9]: Graph of Thought (GoT) prompting enhances ToT by allowing interconnections between thoughts on different branches. It decomposes complex tasks into simpler sub-tasks, solves them independently, and merges the results, thus reducing computational costs.

# 3 Method

## 3.1 MACM Overall Structure

The overall structure of MACM is shown in Figure 2. We have designed an interactive system

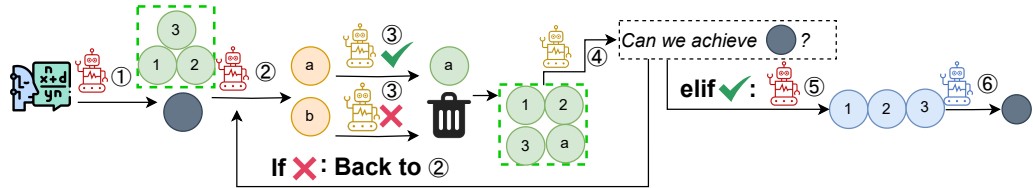

Figure 2: The overall structure of MACM. 📑: Original Math problem; ⌐⌐: Condition list; ✔: True; ✘: False; 🗑: Discard; 🟢: Known Conditions; 🟠: New Conditions; ⚫: Objective; 🤖: Thinker; 🤖: *Judge*; ♟: *Executor*; ①: Initialize the initial condition list and the objective; ②: Explore new Conditions based on current condition list; ③: Check if the new condition is correct; ④: Check if the objective can be achieved based on the current Conditions in the Condition list; ⑤: Designing steps for achieving the objective based on current Conditions; ⑥: Achieve the objective.

comprising three agents: ***Thinker***, ***Judge***, and ***Executor*** to solve complex mathematical problems.

- ***Thinker***: Responsible for generating new thoughts or ideas. This role involves creative thinking and the generation of novel solutions or approaches to problems.
- ***Judge***: Evaluate the thoughts generated by the *Thinker*. It assesses the viability and correctness of new ideas, ensuring that only the most logical and beneficial ones are pursued.
- ***Executor***: Performs calculations or actions according to predefined steps. It is focused on the implementation of the ideas approved by the *Judge*, turning steps into tangible outcomes.

When a mathematical problem is input into our system, the ***Thinker*** initially sets up the *Condition List* and defines the final *Objective* based on the given problem. After initialization, the *Thinker* mines new conditions conducive to the objective from the current *Condition List*, i.e., the *Known Conditions*. The ***Judge*** then assesses these newly mined conditions. If deemed correct, the ***Judge*** incorporates the new condition into the *Condition List*. Otherwise, the new condition is discarded.

Once all new Conditions have been reviewed, we obtain a revised *Condition List*. At this point, the ***Judge*** evaluates whether the current conditions are sufficient to achieve the objective. If the answer is False, the process reverts to step ② for further mining of new conditions. In our experiments, we set a limit of five iterations; if the objective is not met after five rounds of mining, we consider the problem unsolvable. This prevents the program from entering an infinite loop. If the answer is True, the ***Thinker*** designs steps based on the *Known Conditions* to achieve the *Objective*. Finally, the ***Executor*** performs calculations following these steps to produce the final result.

MACM achieves a high level of generalizability by abstracting conditions and objectives from each specific mathematical problem. Through a multi-agent interactive system, where the ***Thinker*** is responsible for ideation and design, the ***Judge*** for inspection and decision-making, and the ***Executor*** for computation, most potential errors in reasoning and calculation are eliminated. By repeatedly mining for conditions and adding the correct ones to the *Condition List*, MACM ensures depth in thinking, making it suitable for analyzing complex mathematical problems.

## 3.2 Theoretical Analysis

MACM moves away from the hierarchical dependencies of previous methods by introducing *Conditions* and *Objectives*. It continuously expands *Known conditions* to derive the final answer, eliminating

the need for manual, problem-specific prompts. This method compresses information from various *Thoughts* into existing *Condition List*, capturing more connections than traditional prompting methods that rely on navigating a hierarchical structure.

The **Thinker** initiates a thought set $\mathcal{T}_1 = \{T_1^1, T_1^2, \ldots, T_1^m\}$ contains $m$ new thoughts from question $Q$ and generates subsequent thought sets $\mathcal{T}_i = \{T_i^1, T_i^2, \ldots, T_i^m\}$ based on the current condition list $\mathcal{C}_i = \{C_1, C_2, \ldots, C_{i-1}\}$. Each condition $C_{i-1}$ is derived from the most accurate thought $T_{i-1}^*$ in $\mathcal{T}_{i-1}$. At each step $i$, the **Judge** selects the correct thought $T_i^*$ in thought set $\mathcal{T}_i$ such that $T_i^* = \arg\max_s P_i^{\text{Judge}}(T_i^*|T_i^s \in \mathcal{T}_i, s \in \{1, \cdots, m\})$, where $P^{\text{Judge}}$ is the probability that the **Judge** confirms the thought as correct. By using this method, we map the whole thoughts space $\mathbf{T}$ to the *Condition List* $\mathcal{C}$. In an ideal situation where $P^{\text{Judge}} \to 1$, we have $\mathbf{T} \to \mathcal{C}$. The probability of arriving at the correct answer $A_{\text{correct}}$ based on the final *Condition list* $\mathcal{C}$ is equal to that based on the entire thoughts space $\mathbf{T}$. Thus we have: $P_{\text{MACM}}(A_{\text{correct}}|\mathcal{C}) = P_{\text{MACM}}(A_{\text{correct}}|\mathbf{T}) > P_{\text{GoT, ToT, CoT}}(A_{\text{correct}} \mid \{T_{ij} \mid i = 1, 2, \ldots, m \text{ and } j = 1, 2, \ldots, n\} \subseteq \mathbf{T})$. In practice, where $P^{\text{Judge}} \not\to 1$, we performed the experiments to test its performance, the results are shown in Section 4.

### 3.3   Using Cases

Our prompts and use cases are shown in Figure 3. It demonstrates the specific process of MACM analyzing algebra and geometry problems. In these two examples, we have employed OpenAI's GPT-4 Turbo [1] as the intelligent agent, which is capable of performing calculations using code. It is endowed with three roles: **Thinker**, **Judge**, and **Executor** by using the following instructions:

---

**For *Thinker***: *You take the role of a **Thinker**. I need you to help me gradually ponder over some problems following my instructions. You need to answer the question by using the following format: Based on Condition A and Condition B, we can get: C.*

**For *Judge***: *You take the role of a **Judge**. I need you to make judgments on some statements. You are only allowed to use the True or False as the final answer.*

**For *Executor***: *You take the role of a **Executor**. I need you to calculate the final result based on the given conditions and steps.*

---

**In the first algebra problem:**

---

*Let $S$ be the set of all real numbers $\alpha$ such that the function $\frac{x^2+5x+\alpha}{x^2+7x-44}$ can be expressed as a quotient of two linear functions. What is the sum of the elements of $S$?*

---

GPT-4 Turbo's raw response reached an incorrect conclusion: $\boxed{x^2 + 5x + \alpha = k(x + 11)(x - 4)}$, which then led to issues in the subsequent code design, ultimately resulting in an incorrect output.

In the MACM analysis process, the **Thinker** initially identifies conditions and objectives from the problem statement and then uncovers new conditions. Although the **Thinker** initially identifies the same incorrect condition as GPT-4 Turbo, the **Judge** detects and rejects this error, preventing its addition to the *Known Conditions*. In the second round, the **Thinker** identifies two new conditions: $\boxed{(-11)^2 + 5 \times (-11) + \alpha_1 = 0}$ and $\boxed{(4)^2 + 5 \times (4) + \alpha_2 = 0}$, which the **Judge** verifies and adds to the *Known Conditions* list. The **Judge** then confirms that the *Known Conditions* are sufficient to achieve the objective. The **Thinker** designs steps to reach the objectives based on these conditions, and finally, the **Executor** performs the necessary calculations to produce the result.

**In the second geometry problem:**

---

*Square $ABCD$ has side lengths of 13 units. Point $E$ lies in the interior of the square such that $AE = 5$ units and $BE = 12$ units. What is the distance from $E$ to side $AD$?*

---

While GPT-4 Turbo's response had the correct theoretical approach, it failed to identify relationships between points in the problem, leading to incorrect expressions and an incorrect final result.

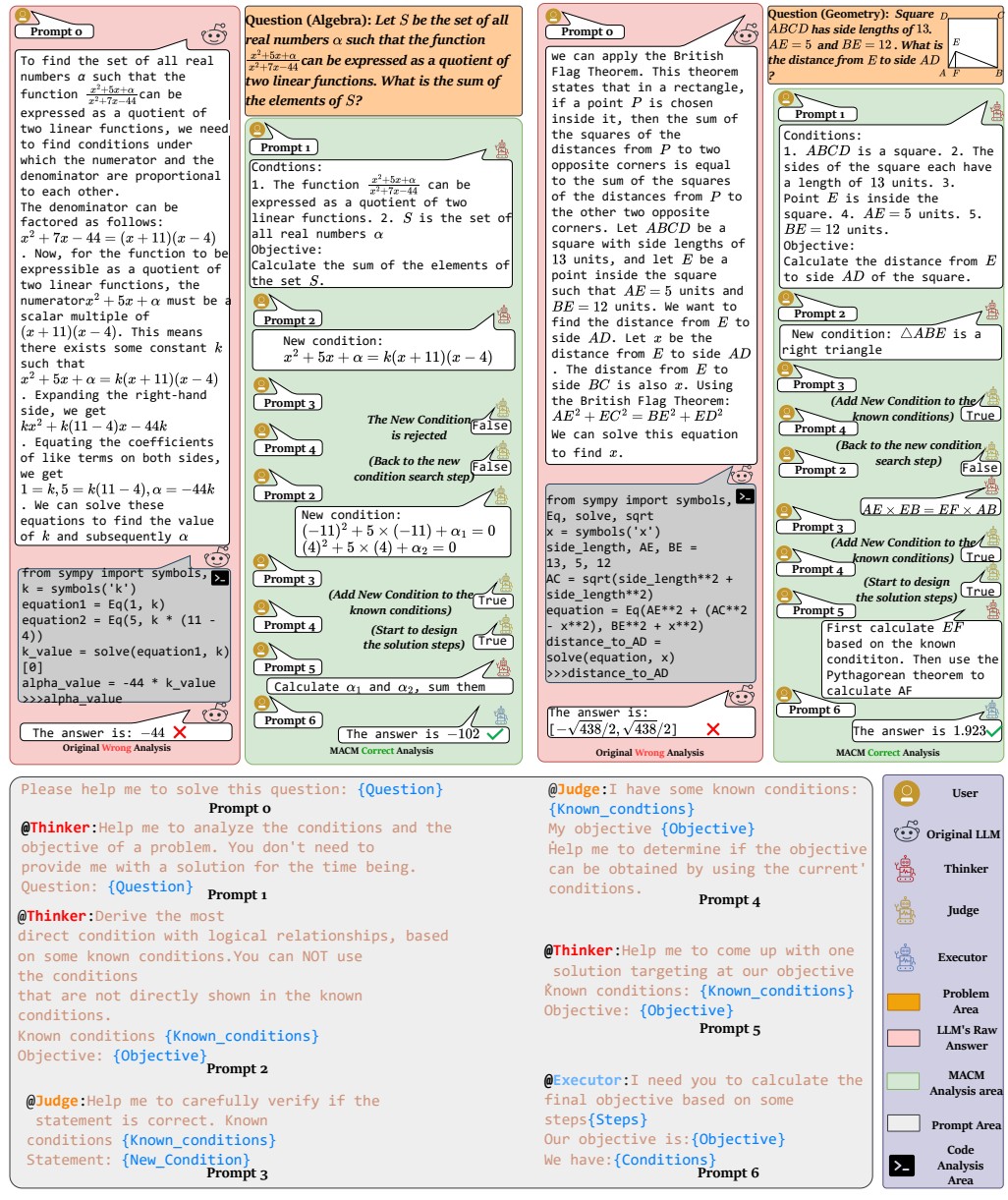

Figure 3: MACM's detailed analysis process for complex mathematical problems with specific prompts, illustrated with an **algebra problem (on the left)** and a **geometry problem (on the right)**. We use one set of prompts that can target different types of problems, with prompts 0-6 displayed in the ☐ below the dialogue box. In these examples, MACM involves three steps: 1. Extracting conditions and the objective. 2. Iteratively identifying new conditions. 3. Solve the problem based on known conditions.

During the MACM analysis, the ***Thinker*** first clarifies the conditions and objectives of the geometry problem and then uncovers new conditions to achieve the goal. Initially, it discovers that: $\boxed{\triangle ABE \text{ is a right triangle}}$. After verification by the ***Judge***, this condition is added to the known conditions. The ***Judge*** then checks if the known conditions are sufficient. A *False* result means more conditions are needed, so the ***Thinker*** continues searching. In the second round, the ***Thinker*** deduces $\boxed{AE \times EB = EF \times AB}$, which the ***Judge*** verifies and adds to the *Condition List*.

Upon confirming sufficiency, the ***Thinker*** plans the steps to solve the problem, and the ***Executor*** performs the calculations to find the result.

In analyzing these two problems, MACM first extracts the specific conditions and objectives from the questions. This allows MACM to directly use these conditions and objectives for prompt design in subsequent processes, enhancing our approach's generalizability. Previous methods like ToT and GoT lack this setup, resulting in poorer generalizability. For example, in the 24-point game experiment with ToT, the lack of this setting necessitated the manual configuration of the following prompt:

> *Evaluate if given numbers can reach 24  \n (sure/likely/impossible)  \n {input}*

In MACM, the $\boxed{\textit{if given numbers can reach 24}}$ is obtained by the first step and the evaluation prompt is generalized to $\boxed{\textit{Evaluate \{objective\} \{input\}}}$, ensuring higher generalizability.

# 4 Experiment

## 4.1 Performance on MATH benchmark

The MATH dataset [6] includes a variety of mathematical problems. It offers seven types of mathematical problems, including geometry, algebra, probability theory, etc., with difficulty levels ranging from 1 to 5. We first tested the overall performance of MACM on the MATH dataset without distinguishing difficulty levels. Afterward, we specifically selected the most difficult mathematical problems from the MATH dataset for testing. The detailed experimental setup is presented in the Appendix B.

Table 1: Accuracy (%) of GPT-4 Turbo on MATH dataset with different prompting methods.

|  | Algebra | Counting and Probability | Geometry | Intermediate Algebra | Number Therogy | Prealgebra | Precalculus | Overall |
|---|---|---|---|---|---|---|---|---|
| I-O | 88.24 | 81.63 | 45.11 | 66.67 | 74.51 | 81.82 | 71.15 | 72.78 |
| CoT | 92.99 | 83.67 | 42.02 | 68.07 | 77.31 | 82.07 | 74.18 | 74.36 |
| SC-CoT | 94.96 | 87.17 | 50.14 | 71.99 | 89.91 | 86.75 | 79.67 | 80.12 |
| CSV [23] | 86.9 | 77.3 | 54.0 | 56.6 | 85.6 | 86.5 | 53.9 | 73.54 |
| CSV + Voting [23] | 95.6 | 89.0 | 64.9 | 74.4 | 94.1 | 91.6 | 67.8 | 84.32 |
| **MACM** | 96.07 | 97.95 | 62.74 | 78.43 | 98.04 | 94.11 | 88.46 | 87.92 |

In Table 1, we compared the accuracy of GPT-4 Turbo on the MATH dataset with various prompting methods. We found that compared to the original GPT-4 Turbo, MACM increased its accuracy by 20%. Compared to CoT, the increase was 13.56%, and compared to SC-CoT, it was 7.8%. Among these, MACM led to the greatest improvement in accuracy for the original GPT-4 Turbo model on number theory problems, at 23.53%. In geometry problems, although MACM has increased the accuracy of GPT-4 Turbo by 17.63%, the final accuracy rate is still only 62.74%. Upon analyzing the causes of errors, we found that many mistakes were due to GPT-4 Turbo's difficulty in accurately understanding the relationships between various geometric figures, thereby failing to design corresponding code to solve the problems. However, in algebra and number theory problems, MACM, by correcting the erroneous analysis of GPT-4 Turbo and helping it explore potential approaches, achieved accuracy rates of 96.07% and 98.04%, respectively. Moreover, compared to the previous SOTA method, CSV prompting, on the MATH dataset, MACM achieves a 3.6 percentage points higher accuracy rate on the same dataset. This demonstrates the effectiveness of MACM in solving mathematical problems.

In Figure 4, we tested the performance of MACM on the **open-source LLaMA series models** on the MATH dataset and compared it with other prompting methods. Since LLaMA models do not have a code interpreter like GPT-4 Turbo, we disabled the code-checking function of MACM in this group of tests. The rest of the experimental setup was consistent with GPT-Turbo. In zero-shot scenarios, the accuracy rates of LLaMA 7B and LLaMA 13B were both below 5% [14]. Majority voting could enhance their accuracies to 6.9% and 8.8% respectively [14], while MACM further increased them to 9.5% and 10.2%. On LLaMA 2 [15] and LLaMA 3 [3], compared to 4-shots, MACM could further improve the accuracy on the MATH dataset by 3-5 percentage points. Overall, We found that MACM can also be applied to LLaMA models, although the performance improvement was not as significant as with GPT-4 Turbo. This is because GPT-4 Turbo has a better understanding of MACM's intrinsic directive prompts, enabling it to find the correct results more effectively.

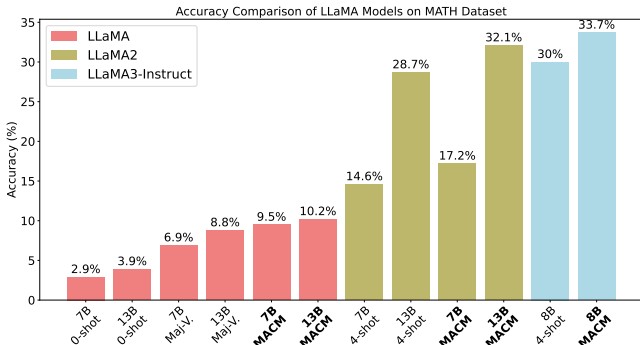

Figure 4: Accuracy comparison of **LLaMA models** on MATH dataset with different methods. Maj-V.: Majority Voting.

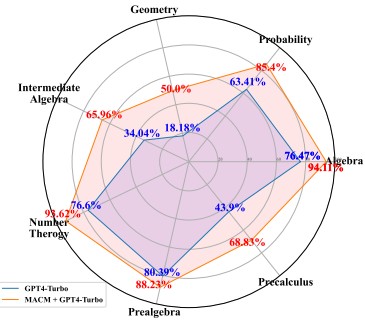

Figure 5: GPT-Turbo's performance on MATH dataset **Level 5** problems with/without MACM.

In Figure 5, We focused on the ability of MACM to solve the **Level 5** mathematics problems in MATH. As shown in the figure, MACM improved the accuracy of GPT-4 Turbo in all seven categories of level 5 problems. The two types of problems that saw the most significant improvement with MACM were the very categories where the original GPT-4 Turbo performed the worst: Geometry and Intermediate Algebra. The original GPT-4 Turbo had an accuracy rate of only 18.18% on Geometry problems and 34.04% on Intermediate Algebra problems. With the support of MACM, it's accuracy rate in Geometry problems increased to 50.0%, and in Intermediate Algebra problems, it increased to 65.96%. This demonstrates MACM's effectiveness in solving difficult mathematical problems.

## 4.2 Comparison with ToT and GoT

Due to the lack of generalization of ToT and GoT prompting methods (See Appendix A for the reason), we were unable to test them on the MATH benchmark. To compare MACM with them, we selected two mathematical problems where their methods are applicable: the 24-point game and sequence sorting. Among these, ToT tested the 24-point game, while GoT studied the sequence sorting problem. The detailed experimental setup is presented in the Appendix B.

Table 2: Accuracy (%) comparison of different methods on various tasks.

| Task | Code Verification | Model | Method | Accuracy (%) |
|---|---|---|---|---|
| 24-points game | ✗ | GPT-4 | IO | 7.3 |
| | ✗ | GPT-4 | CoT [20] | 4 |
| | ✗ | GPT-4 | SC-CoT [20] | 9 |
| | ✗ | GPT-4 | ToT ($b = 1$) [20] | 45 |
| | ✗ | GPT-4 | ToT ($b = 5$) [20] | 74 |
| | ✗ | GPT-3.5 | MACM | 67 |
| | ✗ | GPT-4 | MACM | 91 |
| | ✓ | GPT-4 Turbo | MACM | 99 |
| Sequence sorting (64 elements) | ✗ | GPT-3.5 | GoT [4] | 89.06* |
| | ✗ | GPT-3.5 | MACM | 92 |
| | ✓ | GPT-4 Turbo | MACM | 100 |

In Table 2, We compared MACM with IO, CoT, SC-CoT, and ToT models on the 24-point game. When the model is GPT-4, MACM is 17% higher than ToT ($b = 5$). Note that here, to ensure a fair comparison, we used the standard GPT-4 without any code capabilities. Additionally, with the support of MACM, GPT-3.5 also achieved an accuracy of 67% in the 24-point game, which is higher than the GPT-4 model with ToT ($b = 1$) support. Upon analyzing the reasons for the improvement in accuracy, we found that MACM's Judge corrected many thoughts that were mistakenly evaluated in ToT, leading to GPT-4 choosing incorrect approaches. This correction process significantly contributed to the increase in accuracy. In addition, We compared the GPT-3.5 model's ability to sort 64 numbers using GoT and MACM. MACM outperformed GoT by 2.94%. Note that some results marked with * were estimated from graphs without specific data. Additionally, GPT-4 Turbo achieved 100% accuracy on the Sequence Sorting task due to its problem-based code construction capability.

## 4.3 Performance on other datasets

This section primarily tests two capabilities of MACM:

***1. The ability to solve more challenging mathematical problems.*** We applied MACM to two datasets, SciBench [16] and TheoremQA [5], which claim their difficulty surpasses the middle school level of the MATH dataset, reaching the Undergraduate Level.

***2. Transferability to General Logic Reasoning Tasks.*** Although MACM focuses on solving mathematical problems, to test its applicability, we applied it to the Reclor logic reasoning dataset [21].

Table 3: Accuracy (%) comparison of different methods on the math-related subset of the **SciBench** dataset. Except for MACM, all results are from [16].

| Method | diff (%) | stat (%) | calc (%) |
|---|---|---|---|
| GPT-4 | 32 | 49.33 | 54.76 |
| GPT-4 + CoT | 22 | 50.67 | 42.86 |
| GPT-4 + Few-shots + CoT | 30 | 49.33 | 45.24 |
| GPT-4 + Few-shots + Python | 44 | 68 | 38.1 |
| GPT-4 Turbo | 46 | 61.33 | 52.38 |
| GPT-4 Turbo + CoT | 38 | 64 | 54.76 |
| GPT-4 Turbo + Few-shots + CoT | 32 | 65.33 | 50 |
| GPT-4 Turbo + Few-shots+ Python | 34 | 42.67 | 30.95 |
| GPT-4 + MACM | 48.78 | 69.44 | 64.29 |
| GPT-4 Turbo + MACM | **60.98** | **77.78** | **76.2** |

Table 4: **TheoremQA** results with GPT-4 Turbo.

| Method | Acc. (%) |
|---|---|
| 0-shot | 46.45 |
| CoT | 41.18 |
| PoT | 55.88 |
| MACM | **79.41** |

Table 5: **Reclor** results.

| Method | Acc. (%) |
|---|---|
| 0-shots | 88.1 |
| CoT | 88.5 |
| MACM | **88.9** |

Table 3 displays the testing results of MACM on the SciBench dataset. The SciBench dataset includes problems in chemistry, physics, and mathematics. We only selected the math-related subset for testing, which includes: the *diff*, *stat* and *calc* subset. The experimental setup was consistent with the testing on the MATH dataset (as shown in Appendix B). The results demonstrate that, in contrast to the Chain of Thought (CoT) method, which resulted in decreased accuracy for both GPT-4 and GPT-4 Turbo on this dataset, MACM led to an approximate 20 percentage points.

Table 4 presents the performance of GPT-4 Turbo on the TheoremQA dataset using various prompting methods. The TheoremQA dataset encompasses problems from multiple domains including mathematics, physics, finance, and computer science. Notably, its mathematics subset contains numerous conceptual and definitional questions that do not involve logic reasoning processes; thus, these were not considered in our experiments. We solely tested questions within the TheoremQA mathematics subset that involve logic reasoning and have definite answers (e.g., a specific number, rather than an interpretation of a theorem). The experimental setup was consistent with the testing on the MATH dataset (as detailed in Appendix B). The results indicate that MACM enabled a roughly 30 percentage points increase in accuracy for GPT-4 Turbo on this subset.

Table 5 shows the test results of GPT-4o [13] on the Reclor dataset using various prompting methods. For general logical reasoning problems, we adjusted the computation-related prompts in the original MACM to make them suitable for non-mathematical logical reasoning problems, while maintaining the overall structure consistent. Results show that MACM can also improve the accuracy of general logic reasoning tasks, but the increase is smaller compared to math tasks.

## 4.4 Ablation Study

In this section, we primarily investigate two issues. ① Explore the relationship between MACM Accuracy and LLM Queries, comparing it with other methods. ② Analyze the proportional impact of each component within the MACM on the overall accuracy improvement. We performed these two experiments on 200 randomly selected questions from the MATH dataset that **the original GPT-4 Turbo model answered incorrectly**.

**Trade-off Between Accuracy and LLM Queries:** In general, increasing the number of responses generated by LLMs leads to an improvement in accuracy. Each prompting method has parameters that can increase it, such as the length of the chain $l$ in CoT, the number of voters $v$ in SC-CoT, and the Tree breadth $b$ in ToT. To measure the search efficiency of each method, we compared the relationship between the accuracy and the number of responses generated by GPT-4 Turbo.

We increased the number of answers generated by various prompting methods. For I-O prompting, we directly adjusted the model's response generation parameter $n$, which enables the model to $n$ responses. For CoT, we adjusted not only the parameter $n$ but also the length $l$ of the Chain. For SC-CoT, we built on the first two methods by adding an adjustment to the number of voters $v$.

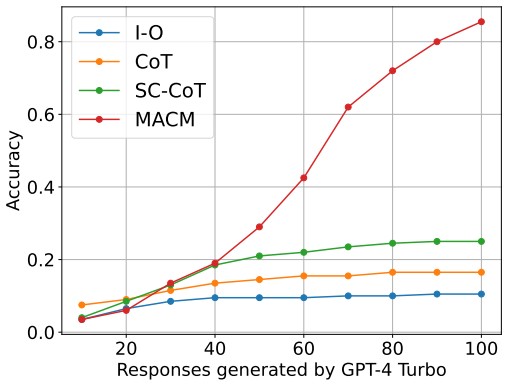

Figure 6: The trade-off between the accuracy and the responses generated by GPT-4 turbo. Compared to I-O, CoT, and SC-CoT, MACM has stronger error correction capabilities when the GPT-4 Turbo generates more responses.

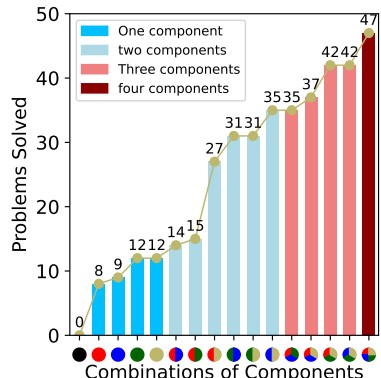

Figure 7: Effectiveness of Component Combinations in MACM. Red: Voting; Blue: Self-Checking; Green: Multi-Agents; Yellow: Condition Mining; A circle containing multiple colors: combinations of different components.

As shown in Figure 6, although I-O, CoT, and SC-CoT only require simple queries to correct the original errors made by GPT-4 Turbo, their upper limits are not high. Even if we continue to increase the number of queries, they can only correct about 20% of the original errors made by GPT-4 Turbo. In contrast, MACM can correct nearly 90% of the original errors of GPT-4 Turbo when the number of queries is high. This is actually quite reasonable because the MACM structure is more complex, including multiple processes of mining conditions and checking. These processes allow the large model to gradually think and identify errors, thus significantly improving accuracy.

**Proportional Impact of Various Components:** To analyze the functions of each component in MACM, we randomly combined the four components within MACM: Condition Mining, Multi-Agent System, Self-Checking, and Voting, and tested their performance. During the experiment, we maintained a maximum of 2 Condition Mining iterations and used 3 Voters.

As shown in Figure 7, the combination of all four components yields the best performance. Among the individual components, Multi-Agents and Condition Mining have comparable error correction capabilities. In the combinations of two components, the pairing of Self-Checking and Condition Mining shows the best performance. Among the three-component combinations, the combination of Multi-Agents, Condition Mining, and either Voting or Self-Checking achieves better results.

## 5 Conclusion

We introduce MACM, a new and generalizable prompting technique that significantly enhances the inferential capabilities of large language models on mathematical problems. MACM can be applied to different types of mathematical questions. Through comparisons in several experiments on the math related datasets, we have verified the superiority of our method over the original prompting methods. With the aid of MACM, the accuracy of the GPT-4 Turbo model on the MATH dataset has increased by 15.14%. Compared to SC-CoT, its accuracy has increased by 7.8%. For the most challenging level 5 mathematical problems in the MATH dataset, its accuracy increased from 54.68% to 76.73%. In the game of 24 points, using the same GPT-4 model, MACM's accuracy is 17% higher than that of ToT. At the same time, by comparing accuracy with the number of times the large model responds, we find that MACM has a higher limit; increasing the number of responses from the large model can significantly improve accuracy. These experiments demonstrate MACM's generalizability and its powerful error-correction capability for complex mathematical problems in original LLMs.

## Acknowledgement

This work was supported in part by a research grant from Cisco Research and by an Amazon Research Award.

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

## A    Why is the generalizability of ToT and GoT limited

This section demonstrates specific examples of using ToT and GoT to further illustrate why their generalizability is limited.

ToT conducted three sets of experiments in the original study, requiring specially designed prompts for each. The official implementation on their GitHub page https://github.com/princeton-nlp/tree-of-thought-llm/tree/master/src/tot/prompts includes the specific prompts set up for each experiment. Taking the 24-point game as an example, specific prompts such as propose prompt, value prompt, and value last step prompt were required (lines 51 to 134 in game24.py). During ToT's operation, the LLM executes traversal searches, voting, filtering, etc., based on these written prompts. The authors also mention in the ToT readme section *How to Add a New Task* (https://github.com/princeton-nlp/tree-of-thought-llm?tab=readme-ov-file#how-to-add-a-new-task) that setting up task-specific prompts is necessary for different problems, further illustrating the limited generalizability of ToT and GoT due to the need for task-specific prompt engineering.

GoT faces the same issue, with their original paper conducting experiments in four tasks: Sorting, Set Operations, Keyword Counting, and Document Merging. For each type of problem, specific prompts must be set up on their official GitHub. Taking Sorting as an example, the specific prompts for sorting are displayed in https://github.com/spcl/graph-of-thoughts/blob/main/examples/sorting/example_prompts_sorting_032.md. They provide the LLM with the instruction: *Split the following list of 32 numbers into 2 lists of 16 numbers each, the first list should contain the first 16 numbers and the second list the second 16 numbers. Only output the final 2 lists in the following format without any additional text or thoughts!* This instruction is clearly tailored to this specific problem, illustrating the limited generalizability of GoT due to the necessity for problem-specific prompt engineering, similar to ToT.

Take ToT as an example, they tested three tasks, for the game of 24 task, the propose prompt is:

> *Input: 2 8 8 14 \n Possible next steps: \n 2 + 8 = 10 (left: 8 10 14) \n 8 / 2 = 4 (left: 4 8 14) \n 14 + 2 = 16 (left: 8 8 16) \n 2 * 8 = 16 (left: 8 14 16) \n 8 - 2 = 6 (left: 6 8 14) \n 14 - 8 = 6 (left: 2 6 8) \n 14 / 2 = 7 (left: 7 8 8) \n 14 - 2 = 12 (left: 8 8 12) \n Input: {input} \n Possible next steps:*

for the cross word task, the propose prompt is:

> *Let's play a 5 x 5 mini crossword, where each word should have exactly 5 letters. \n {input} \n Given the current status, list all possible answers for unfilled or changed words, and your confidence levels (certain/high/medium/low), using the format "h1. apple (medium)". Use "certain" cautiously and only when you are 100% sure this is the correct word. You can list more than one possible answer for each word.*

for the creative writing task, the propose prompt is:

> *Write a coherent passage of 4 short paragraphs. The end sentence of each paragraph must be: {input} \n Make a plan then write. Your output should be of the following format: \n Plan: \n Your plan here. Passage: \n Your passage here.*

Each time the problem is changed, both ToT and GoT require an update to their respective prompts. The requirement to tailor prompts for each specific problem limits the generalizability of ToT and GoT to broader issues. MACM successfully addresses this challenge.

## B    Experimental Setup

**For the experiments on the MATH dataset:** We utilized the GPT-4 Turbo model (between January 1, 2024, and February 1, 2024) to test MACM's performance on the MATH dataset. For tests that did not distinguish by difficulty, we randomly selected one-third of the questions from the MATH dataset

for evaluation. For the high-difficulty tests, we extracted all questions with a difficulty level of 5 and randomly selected half of the questions from each category for testing. The experiment are performed by using I-O, CoT, SC-CoT, and MACM methodologies. For all prompting methods, we standardized the number of responses $n$ generated by GPT-4 Turbo to 1, $Top_k = 1$ , and the temperature $t = 0$. For CoT, we set the maximum length of the chain $l = 5$, for SC-CoT, the number of voters $v = 5$, and the maximum length of the chain $l = 5$. For these three methods, we consistently maintained max_tokens at 512. For MACM, we kept the thinker's max_tokens at 512, the judge's max_tokens at 4, and the executor's max_tokens at 256.

**For the 24-point game experiment**: We sourced data from 4nums.com, which offers 1,362 games ranked from easy to hard based on the time it takes humans to solve them. We focused on a subset of these games, specifically those ranked 901 to 1,000 (The same as ToT), to test on relatively difficult challenges. Success for each task is defined as producing a valid equation that results in 24, utilizing each of the input numbers exactly once. The performance metric is the success rate across these 100 challenging games.We utilized the GPT-4 and GPT-3.5 model (between January 1, 2024, and February 1, 2024) to perform the experiments. The MACM configuration for this experiment includes setting the number of responses generated by the model $n = 1$, $Top_k = 1$, temperature $t = 0$, with the thinker's max_tokens at 512, the judge's max_tokens at 4, and the executor's max_tokens at 256.

**For the sequence sorting experiment**: We randomly generated 100 sequences, each containing 64 elements, for testing. We utilized the GPT-3.5 model (between January 1, 2024, and February 1, 2024) to perform the experiments. The MACM configuration for this experiment includes setting the number of responses generated by the model $n = 1$, $Top_k = 1$, temperature $t = 0$, with the thinker's max_tokens at 512, the judge's max_tokens at 4, and the executor's max_tokens at 256.

## C  Limitation and Discussion

While MACM significantly enhances the accuracy of large language models in tackling complex mathematical challenges, it incurs the cost of multiple invocations of the large language model for inference, leading to increased problem-solving time. Additionally, our evaluations using the MATH dataset indicate limitations in effectively addressing geometry problems. Addressing these challenges necessitates further advancements in the model's own cognitive capabilities. A proposed strategy involves employing prompting methods like MACM to assist the LLM in eliminating incorrect responses. This approach enables the creation of expansive, high-quality datasets, which are otherwise challenging to compile manually, and subsequently refining the LLM with these datasets. Through this iterative process, the model's intrinsic intelligence is progressively augmented. This research direction will constitute our future work.

