# OpenReview forum: "MACM: Utilizing a Multi-Agent System for Condition Mining in Solving Complex Mathematical Problems"
_NeurIPS.cc/2024/Conference — NeurIPS 2024 poster_

### Official Review · Reviewer_upb5 · 2024-07-11

**Soundness:** 3
**Presentation:** 3
**Contribution:** 2
**Rating:** 7
**Confidence:** 5

**Summary:**

This paper aims to address the limitations of existing methods that require distinct prompt designs for different mathematical problems. The authors propose a general Multi-Agent System for conditional Mining (MACM) method that uses three LLM agents (Thinker, Judge, Executor) to iteratively propose the conditions for a problem, verify whether the existing conditions can reach the objective, and execute calculations. Experimental results clearly show the accuracy improvements of MACM.

**Strengths:**

The main strengths include:

1.	The motivation to design a method that does not require prompt design for different mathematical problems is important and makes sense.
2.	The authors clearly explain the details of MACM, which makes it easy to follow.
3.	The authors conduct experiments on various datasets, and the accuracy improvements are significant.

**Weaknesses:**

The main weaknesses include:

1.	The essential technical contributions of this paper may be limited, and an explanation is needed as to why the method is effective. (please refer to question 1 below).
2.	The theoretical analysis is a little weird and needs further explanations. (please refer to question 2 below).
3.	More experimental results are needed to demonstrate the effectiveness of the proposed method. (please refer to questions 3-4 below).

**Questions:**

1.	In the first using case in Figure 3, why do the LLM gets two different “New condition” (one correct, one incorrect) under the same prompt (prompt 2)? Does this mean that thinker in this paper is more likely to take advantage of the randomness of large models in repeated condition generation, rather than really improving the ability of LLMs to discover new conditions?
2.	In theoretical analysis, how to ensure that the thought space T can be traversed, i.e., why P_MACM(A_correct|C) = P_MACM(A_correct|T)?
3.	In Section 4.2, it's necessary to rerun the results of GPT4-ToT instead of copying the results from the original paper, because the capability of GPT4 is also improving. It is difficult to tell whether the improvement in Table 2 is caused by improvements in GPT4 itself or by the method in this paper.
4.	In Section 4.4, I think the most important measure of efficiency for applying this method is to compare the number of API calls from this method with different prompt methods (e.g., ToT, GoT) when obtaining the results in Tables 1 and 2. This is the most direct indicator of the cost in practice.

**Limitations:**

Yes, the authors have adequately addressed the limitations.

---

> ### Author Rebuttal · Authors · 2024-08-07
>
> Thank you for reviewing our work. Here are our responses to your concerns:
>
> ---
>
> **For *Weakness 1* and *Question 1***
>
> > why do the LLM gets two different “New condition” (one correct, one incorrect) under the same prompt (prompt 2)?
>
> Thanks for the question! In MACM, the conversation takes place in **multiple rounds**, with previous dialogue information added to the conversation history. This means that in Figure 3, after the first prompt 2 generates an incorrect new condition, the judge's evaluation of this new condition will also be recorded in the conversation history (the leftmost **prompt 3** and **prompt 4** in **Figure 3**). This reduces the likelihood of the model considering incorrect conditions in subsequent sampling processes.
>
> > Does this mean that thinker in this paper is more likely to take advantage of the randomness of large models in repeated condition generation, rather than really improving the ability of LLMs to discover new conditions?
>
> MACM includes various thought-guiding processes that help the LLM continuously correct potential past errors and guide the large model to gradually use existing conditions to obtain the correct new conditions, ultimately finding the correct result. In addition, In the process of the judge evaluating each new condition, we have configured the use of a code interpreter. This can significantly reduce potential errors and gradually guide the model towards producing correct answers.
>
> **For *Weakness 2* and *Question 2***
>
> In the theoretical analysis section, our aim is to highlight the advantages of MACM over ToT and GoT in **ideal situations**. Specifically, each condition search in MACM can be viewed as a new start, meaning it doesn't need to begin its search from a few initial paths like ToT and GoT. Therefore, theoretically, as the search time increases, MACM's exploration space will gradually approach the thought space  $T$
>
> In contrast, ToT and GoT are constrained to specific trajectories, making it difficult for them to consider alternative perspectives. Thus, MACM offers greater flexibility and potential in exploring and traversing the thought space, which can enhance problem-solving effectiveness.
>
> **For *Question 3***
>
> Thank you for your suggestion. We have standardized our testing by using GPT-4-1106-preview to re-evaluate the results for the 24-point game, with all code-interpreter functionalities disabled during the testing process. The testing code for IO/CoT/ToT is sourced from ToT's official GitHub repository.
>
> | Method                | Accuracy (%) |
> |:----------------------:|:------------:|
> |  IO               |      $23$    |
> |  CoT |      $25$       |
> | SC-CoT  |      $34$      |
> | ToT (b = 1)|     $55$       |
> | ToT (b = 5)|     $78$      |
> | MACM           |     $91$      |
>
> Experimental results show that MACM still outperforms ToT. By analyzing the error cases in ToT, we found that many problems that MACM can solve but ToT cannot are due to MACM's judge correcting these errors.
>
> **For *Question 4***
>
> Thanks for your suggestion!
>
> Regarding the average number of responses $n$ generated by LLM, we have summarized the results in the following table.
>
> For Table.1
>
> | Method         | Average responses $n$| Accuracy (%) |
> |:----------------:|:-----------------:|:------------:|
> | CoT            | $5$               | $74.36$|
> | SC-CoT         | $25$              |$80.12$|
> | CSV†     | $7.552$                 | $73.54$|
> | CSV-Voting†     | $47.26$          | $84.32$|
> | MACM           | $40.48$           | $87.92$|
>
> †: They did not provide official code implementation and the related data is not involved in their paper. The data come from our own implementation.
>
> **For Table.2 24-point problem**:
>
> | Method  | Average responses $n$ | Accuracy (%) |
> |:---------:|:---------------------------:|:------------:|
> |     CoT*          |             $4.17$  |  $25$       |
> |     ToT ($b=1$)*  |          $23.72$     |     $55$       |
> | ToT ($b=5$)* | $67.81$                   |    $78$       |
> | MACM (w/o code verification)  | $63.3$  |     $91$      |
> | MACM (w code verification)    | $22.12$ | $99$      |
>
> **For Table.2 sequence sorting**:
> | Method  | Average responses $n$|  Accuracy (%) |
> |:---------:|:---------------------------:|:------------:|
> |     GoT*  |  $54.3$                  |     $89.06$       |
> | MACM (w/o code verification) | $51.72$|      $92$       |
> | MACM (w code verification)   | $6.882$ |      $100$       |
>
> *: The related data is not involved in their paper, we obtained the data from their official GitHub repository.
>
> Experimental results show that, compared to baselines such as CSV, ToT, and GoT, MACM requires fewer LLM responses even without a code interpreter function, while achieving higher accuracy.
>
> ---
>
> Thank you again for your feedback! We hope our response can address your concerns.

---

> > ### Comment · Reviewer_upb5 · 2024-08-13
> > **Response to authors' rebuttal**
> >
> > Thanks for the authors' clarifications. The authors have addressed most of my questions. I am pleased the authors could add the GPT4 experiments and efficiency comparison, and the results generally enhance the work. I would like to increase my score. I hope our discussions can be included in the further version.

---

### Official Review · Reviewer_pd1k · 2024-07-12

**Soundness:** 1
**Presentation:** 2
**Contribution:** 1
**Rating:** 4
**Confidence:** 4

**Summary:**

The paper proposes a prompting method called Multi-Agent System for conditional Mining (MACM). MACM involves three agents, Thinker, Judge, and Executor, who maintain a condition list and try to solve the problems when the conditions are sufficient.

The paper conducts experiments with GPT and Llama series on MATH, 24-points game and sequece sorting. The results show MACM’s superiority to prompting methods like ToT and GoT under these settings.

**Strengths:**

The paper does show significant improvement of models’ performance with MACM unber specific settings.

**Weaknesses:**

- MACM seems to lack novelty. MACM’s idea of maintaining a more free-form context for planning has been explored by previous works like Cumulative Reasoning.
- The experimental setups in the paper seem confused and lack rigor, making the results less convincing. Examples include but are not limited to:
  - In section 4.2, It is unreasonable to state that ToT and GoT are incompatible with MATH and quit comparing with them. Algorithms like ToT have been applied to open-form QA tasks by previous works like LLM reasoners.
  - Across most experiments, it is confusing to distinguish between 0-shot/IO and CoT for GPT-4 on reasoning tasks, for GPT-4 seems to always use CoT on reasoning tasks if not specially prompted.
  - Especially, experimental setups described in Appendix B seem rather unreasonable:
    - The paper states that it always uses “$Top_{k}=1$, and the temperature $t=0$”, i.e. greedy decoding, but this seems incompatible with SC-CoT (though still possible with minor randomness existing).
    - The meaning of “length of the chain $l$” in CoT seems not explained across the paper.
    - The `max_tokens` seems too small for complex tasks like MATH.
- The writing lacks clarity and is hard to follow:
  - The paper tends to list many settings that are different in multiple dimensions and their results in a line (e.g. Figure4,7, Table2,3), making attribution of the results difficult.
  - The paper spends much space on specific examples (\~2 pages across pages 4-6) but seems to show few special things.

Hopefully, the authors could put more work into experimental setup and presentation to improve the quality of the work.

# **After rebuttal and discussion**

The discussion period brings up

- 2 new weaknesses:
    1. According to Figure 4, MACM's gains on open-source models are not clear and may be marginal. Could you compare MACM with 0-shot and Majority voting, at least on LLaMA3-Instruct 8B?
    2. To test the impact of different models on MACM as **a new method**, it is not sufficient for GPT-3.5 to only test with MACM -- comparisons with baselines are needed. However, this could also be achieved with open-source models, which should be much cheaper and faster.
- 1 new question: The exact versions of GPTs in the paper are unclear. You should at least annotate the exact versions and better compare the methods with the exact same version.

However, most concerns have been resolved by the authors in their replies.

Considering the improvement in the rebuttal and discussion, I believe that the submission has a contribution for **proposing a new effective and general prompting method, which is limited in being only applicable to strong models**.

However, the submitted manuscript is indeed not good enough because it **lacks many details and is not well-formed enough, actually bringing an unnecessary burden for the community to follow the work**, as shown in the long discussion. I tend to take it as not ready for publication in its current form. Hopefully, the authors will add all the necessary content to future versions.

In summary, I would like to keep my final rating as 4 as a reminder of the presentation problem and leave it to AC to decide whether this submission should be accepted.

**Questions:**

See above in weaknesses.

**Limitations:**

See above in weaknesses.

---

> ### Author Rebuttal · Authors · 2024-08-07
>
> Thank you for reviewing our work. Here are our responses to your concerns:
>
> ---
>
> **For *Weakness 1:***
>
> MACM and Cumulative Reasoning differ significantly, especially in the rechecking and voting processes. Intuitively, Cumulative Reasoning has an accuracy of only $72.2$% on MATH. According to OpenAI's official data [1] , GPT-4 Turbo itself can achieve an accuracy of $72.6$% on math problems. Meanwhile, MACM achieves an accuracy of $87.92$% on MATH, which is significantly higher than Cumulative Reasoning.
>
> **For *Weakness 2 (a):***
>
> > In section 4.2, It is unreasonable to state that ToT and GoT are incompatible with MATH and quit comparing with them.
>
> ToT and GoT have significant limitations in practical applications. They can only be applied to data with relatively fixed formats and low information content, as stated in our **Appendix A**. In contrast, MACM has greater versatility and applicability. In LLM reasoners, ToT has only been applied to relatively simple tasks like GSM8K, AQuA, and Game24.
>
> To our knowledge, there has not yet been any work that applies ToT to an entire MATH dataset.
>
> **For *Weakness 2 (b):***
>
> > for GPT-4 seems to always use CoT on reasoning tasks
>
> Currently, there is no evidence to suggest that GPT-4 **always** uses CoT on reasoning tasks. To ensure a comprehensive and accurate comparison, we adopted the same experimental settings as previous baselines, including CSV, ToT, and GoT prompting methods. This involved thoroughly testing both IO and CoT methods.
>
> **For *Weakness 2 (c):***
>
> > experimental setups described in Appendix B seem rather unreasonable
>
> 1. The specific process for performing **SC-CoT** using **greedy decoding** is as follows: First, let LLM generate several plans using a prompt such as, `Please help me to list several different plans for solving this problem.` Then, add each of these plans to different execution paths and continue the subsequent reasoning separately. Finally, aggregate the results and conduct a vote. We use greedy decoding here to reduce the randomness in the experiments.
>
> 2. The length of the chain $l$ refers to the number of reasoning steps the model performs in CoT. We set this parameter to fix the number of reasoning steps to facilitate the evaluation of efficiency.
>
> 3. For MACM, we set the thinker's `max_tokens` to $512$, the judge's max_tokens to $4$, and the executor's `max_tokens` to $256$. These `max_tokens` values are for each step of the process. MACM decomposes each problem in the MATH dataset into multiple reasoning and judgment steps, making these `max_tokens` settings reasonable and feasible for each step.
>
> **For *Weakness 3 (a):***
>
> > The paper tends to list many settings that are different in multiple dimensions and their results in a line (e.g. Figure4,7, Table2,3), making attribution of the results difficult.
>
> Different models and prompting methods can affect the final results.
>
> In **Figure 4**, we tested various LLaMA models using different prompting methods on the MATH dataset. Different model versions are distinguished by different colors, with the model parameters labeled on the first row of the x-axis and the prompting methods labeled on the second row of the x-axis.
>
> In **Figure 7**, we tested the contribution of four different components in MACM to the improvement in model accuracy. Different components are indicated by different colors on the x-axis, and combinations of different numbers of components are represented by bars of various colors.
>
> In **Table 2**, we have 5 columns: the first column lists the tasks tested, the second column indicates whether a code interpreter was used for verification, the third column specifies the model used, the fourth column details the prompting method employed, and the fifth column shows the resulting accuracy.
>
> In **Table 3**, we have 4 columns: the first column lists the methods used, which include information about the model, prompting method, and whether Python was used. The second to fourth columns correspond to three different subsets of SciBench.
>
> **For *Weakness 3 (b):***
>
> > The paper spends much space on specific examples (~2 pages across pages 4-6) but seems to show few special things.
>
> As with previous prompting papers, we have listed specific examples to help readers better understand the details of our method. For instance, in the Graph of Thought [2], refer to Figures 2, 3, 4, and pages 3, 5, 6. Similarly, in the CSV paper [3], refer to Figures 1, 3, 4, and pages 3, 5, 6, 8.
>
> **Reference**
>
> [1] GPT-4o text evaluation report.
>
> [2] Graph of Thoughts: Solving Elaborate Problems with Large Language Models, AAAI 2024
>
> [3] Solving Challenging Math Word Problems Using GPT-4 Code Interpreter with Code-based Self-Verification, ICLR 2024
>
> ---
>
> Thank you again for your feedback! We hope our response can address your concerns.

---

> > ### Comment · Reviewer_pd1k · 2024-08-08
> >
> > Thanks a lot for the effort you put in your rebuttal, which clear up some of my doubts. However, below are my follow-up comments:
> >
> > **For *Weakness 1:***
> >
> > 1. Could you detail the difference between condition mining / claim proposal, rechecking / verification and  judge / reporter in MACM / CR (Cumulative Reasoning) respectively?
> > 2. Where is the implementation of the voting in MACM detailed? I am sorry to be only able to find a number of voters as 3 in lin 259.
> > 3. The comparison between MACM and CR might be unfair for using different versions of GPT-4 (CR uses `gpt-4-1106-preview`). I notice the time span in lines 369-370 but I want to check which exact version you use. Isi it `gpt-4-0125-preview`?
> > 4. It would be best if you could compare with CR on complex tasks like MATH/SciBench/TheoremQA.
> >
> > **For *Weakness 2 (a):*** The algorithms of ToT and GoT are fundamental to compare with on complex tasks and should be applicable to tasks like MATH. SSC-CoT [1] implements ToT for MATH. And GSM8K & AQuA are quite similar to MATH in format.
> >
> > **For *Weakness 2 (c):***
> >
> > 2. How you decompose the chain into steps is unclear. As suggested in ToT, "the decomposition of thoughts (e.g. is each zi a phrase, a sentence, or a paragraph) is left ambiguous".
> >
> > **For *Weakness 3 (a):***  For results related to **multiple independent variables (e.g. model, prompting, code usage, task)**, I recommend dividing them into different dimensions/groups in a table or different tables/figures, which allows **easier comparison in a controlled variable way**.
> >
> > **For *Weakness 3 (b):*** It is good to provide detailed examples. But Section 3.3 seems highly overlapping with Figure 3, bringing little new information.
> >
> > **New questions:**
> >
> > 1. The exact versions of GPTs in the paper are unclear. You should at least annotate the exact versions and better compare the methods with the exact same version.
> >
> > For the improvement so far, the scores in my mind should be:
> >
> > Soundness: 2
> > Presentation: 2
> > Contribution: 2
> > Rating: 4
> >
> > I will keep updating the scores as the rebuttal improves.
> >
> > Again, thanks for your work and efforts!
> >
> > [1] Zhao, Zilong, et al. "Stepwise Self-Consistent Mathematical Reasoning with Large Language Models." arXiv preprint arXiv:2402.17786 (2024).

---

> > > ### Author Response · Authors · 2024-08-09
> > > **Reply to reviewer pd1k (Part 1)**
> > >
> > > Thank you for your reply!
> > >
> > > ---
> > >
> > > **For *Weakness 1 (1)***:
> > >
> > > > Could you detail the difference between condition mining / claim proposal, rechecking / verification and judge / reporter in MACM / CR (Cumulative Reasoning) respectively?
> > >
> > > ***For condition mining / claim proposal***
> > >
> > > In the first step of the **CR**, *Preliminary Contents* and *Hints* will be proposed based on the topic. These conditions will serve as the foundation for determining the subsequent steps in the later claim proposal. Details are shown in their official github: `cumulative-reasoning/CR-Agent/prompts/cr/math.md` file.
> > >
> > > Structurally, the claim proposal of CR is as follows:
> > >
> > > ***problem*** --> ***(condition 1,...condition n)*** --> ***step 1*** --> ***step 2***... ***step n***--> ***solution***
> > >
> > > In **MACM**, **condition mining is an ongoing process**. We do not set Preliminary Contents and Hints. In the first step, the agent needs to perform thorough condition mining for the given problem. All conditions verified to be correct will be added to our condition list. This means we **diminish the hierarchical progression** among these conditions and strive to keep them on the same level. **At this stage, we do NOT consider the specific path** to solving the particular problem but focus on **thoroughly exploring specific conditions** that might be helpful in solving the problem.
> > >
> > > Structurally, the condition mining of MACM is as follows:
> > >
> > > ***problem*** --> ***(condition 1,...condition n)*** --> ***(condition 1,...condition n+1)*** --> ...***(condition 1,...condition n+m)***
> > >
> > > **Note**: At this step, MACM has not yet started designing the specific steps to solve the problem.
> > >
> > > ***For rechecking / verification***
> > >
> > > In **CR**, the main purpose of verification is to precisely check the computational process and logical derivation, as indicated in the last sentence of the second paragraph of Section 3.1 in their paper: *verifiers translate these proposals into formal systems for validation, employing either symbolic reasoning systems or incorporating a code environment*.
> > >
> > > In **MACM**, the rechecking process involves not only **using a code interpreter for strict mathematical calculations** but also **leveraging LLM for some intuitive judgments**. For example, before adding a condition obtained during the condition mining process to the condition list, we need to determine **whether it is helpful in solving the target problem**. This is a problem that is difficult to precisely judge using strict calculations. Therefore, we hope that during the rechecking process, the LLM can help us recheck these newly generated conditions, not only verifying their mathematical accuracy using the code interpreter but also subjectively judging whether we really need this condition.
> > >
> > > ***For judge / reporter***
> > >
> > > In **CR**, the role of the reporter is to report the final results (e.g., whether the hypothesis is correct, the specific process of the 24-point calculation, etc.), as described in Appendix A *Appendix for Prompts* of their paper.
> > >
> > > Structurally, the reporter of CR will be responsible for this step:
> > >
> > > ***step n***--> ***solution***
> > >
> > > In MACM, one of the important tasks of the Judge is to determine whether the current condition list is sufficient to **design a path** that can lead to the final result.
> > >
> > > Structurally, the Judge of MACM will be responsible for this step:
> > >
> > > ***(condition 1,...condition n+m)***-->***(step 1,...step n)***
> > >
> > > **Summary**： Compared to CR, MACM conducts condition mining in a broader thought space, which is helpful for solving relatively complex problems with a wide range of thinking (for example, solving certain problems that require the integration of knowledge from different fields, such as combining algebra and geometry)
> > >
> > > **For *Weakness 1 (2)***:
> > >
> > > Based on our experimental observations, voting for more complex agents can effectively improve their overall accuracy.
> > >
> > > The specific implementation of this can be found in supplementary material code in `main.py` at `line 74` (voting starts) and `lines 114-115` (voting summary).
> > >
> > > To save on compute cost, we have adopted a method of voting directly on the overall result, rather than voting on each intermediate step as done in ToT.
> > >
> > > **For *Weakness 1 (3)***:
> > >
> > > In our experiments, the model we used is the same as in CR, which is `gpt-4-1106-preview`. This can be found in our supplementary material code in `utils/gpt_robots.py`. Thank you for your suggestion, we will clearly indicate this information in future versions of the paper.

---

> > > > ### Author Response · Authors · 2024-08-09
> > > > **Reply to reviewer pd1k (Part 2)**
> > > >
> > > > **For *Weakness 1 (4)***:
> > > >
> > > > We compared the performance of CR and MACM on the MATH and SciBench datasets and will update the corresponding experimental results in a future version of the paper. The details are as follows:
> > > >
> > > > **MATH:**
> > > >
> > > > Due to experimental funding limitations, we directly collected the data measured by CR on the MATH dataset from the original CR paper. The comparison results are as follows:
> > > >
> > > > | Method         | Algebra | Probability | Geometry | InterAlgebra | NumTheory | Prealgebra | Precalculus | Overall |
> > > > |:----------------:|:---------:|:--------------------------:|:----------:|:----------------------:|:---------------:|:------------:|:-------------:|:---------:|
> > > > | CR       | $86.3$ | $71.1$              | $53.7$|$51.5$            | $88.7$     | $86.6$   | $51.8$   |$72.2$ |
> > > > | MACM       | $96.07$ | $97.95$               | $62.74$ | $78.43$            | $98.04$    | $94.11$  | $88.46$   | $87.92$ |
> > > >
> > > > **SciBench:**
> > > >
> > > > ***Experiment Setup***
> > > >
> > > > We used the code from CR's official GitHub to test SciBench. The testing code is located at `cumulative-reasoning/CR-Agent/infer/inference.py`. The testing format prompt was based on the prompt they used for testing the MATH dataset, which can be found in `cumulative-reasoning/CR-Agent/prompts/cr/math.md`. The `model_name_or_path` is set to `gpt-4-1106-preview`. All other parameters were kept at their default values. As in our paper, we tested its performance on the diff/stat/calculus subsets of SciBench. Some CR results are approximately equal to the correct result, we also treat these as correct. For example, if CR's final output is *approximately 0.336*, but the ground truth is $0.3359$, we treat this as correct. the comparison results are as follows.
> > > >
> > > > ***Experiment Results***
> > > >
> > > > | Method         | diff (%)| stat (%)| calc (%)|
> > > > |:----------------:|:---------:|:--------------------------:|:----------:|
> > > > | CR       | $56.36$ | $61.05$              | $59.61$|
> > > > | MACM       | $60.98$ | $77.78$               | $76.2$ |
> > > >
> > > > ***Experiment Conclusion***
> > > >
> > > > MACM's accuracy on the MATH dataset and the diff/stat/calculus subsets of SciBench is higher than CR.
> > > >
> > > > **For *Weakness 2 (a)***:
> > > >
> > > > ***ToT Comparison***
> > > >
> > > > ***Experiment Setup***
> > > >
> > > > In SSC-CoT **Section 5.1, baseline part, line 5**, they mention that the model they used is `gpt-3.5`. Since they did not specify the exact version of `gpt-3.5` used, we directly designated the model name as `gpt-3.5` for comparison. Additionally, SSC-CoT authors did not mention using a Code Interpreter for ToT experiments, so we disabled the code interpreter function in MACM for this comparison.  The result on Level 5 problems (1324 cases) is shown below.
> > > >
> > > > ***Experiment Results***
> > > >
> > > > | Method         | Algebra | Probability | Geometry | InterAlgebra | NumTheory | Prealgebra | Precalculus | Overall |
> > > > |:----------------:|:---------:|:--------------------------:|:----------:|:----------------------:|:---------------:|:------------:|:-------------:|:---------:|
> > > > | ToT      | $27.4$ | $14.6$         | $3.8$|$3.2$          | $11.0$     | $34.7$   | $1.5$   |$15.3$ |
> > > > | MACM       | $32.9$ | $18.7$    | $5.3$ | $4.6$       | $18.2$    | $54.9$  | $4.4$   | $21.4$ |
> > > >
> > > > ***Experiment Conclusion***
> > > >
> > > > On more challenging logical reasoning problems, such as level 5 problems in the MATH dataset, MACM performs better compared to ToT.
> > > >
> > > > ***GoT Comparison***
> > > >
> > > > According to **SSC-CoT Section 5.1, baseline part, lines 6-12**, GoT is not tailored specifically for mathematical reasoning, which makes it difficult for us to compare it on the MATH dataset.
> > > >
> > > > **For *Weakness 2 (c) 2***:
> > > >
> > > > We used the following prompt to decompose the chain into steps:
> > > >
> > > > `Please help me design a plan to solve the following problem: {problem}. Your plan should be within 5 steps`

---

> > > > > ### Author Response · Authors · 2024-08-09
> > > > > **Reply to reviewer pd1k (Part 3)**
> > > > >
> > > > > **For *Weakness 3 (a)***:
> > > > >
> > > > > Thank you for your suggestion! We will break down the tables/figures containing multiple variables into a more easily comparable format. Taking our **Figure 4** as an example, we will split the different versions and parameters of the LLaMA models into multiple tables, as shown below:
> > > > >
> > > > > Table 1: Accuracy comparison of **LLaMA 7B**
> > > > >
> > > > > | Method                    | Acc. (%)         |
> > > > > |:---------------------------:|:--------------------:|
> > > > > |0-shot|$2.9$%|
> > > > > |Majority Voting|$6.9$%|
> > > > > |MACM|$9.5$%|
> > > > >
> > > > > Table 2: Accuracy comparison of **LLaMA 13B**
> > > > >
> > > > > | Method                    | Acc. (%)         |
> > > > > |:---------------------------:|:--------------------:|
> > > > > |0-shot|$3.9$%|
> > > > > |Majority Voting|$8.8$%|
> > > > > |MACM|$10.2$%|
> > > > >
> > > > > Table 3: Accuracy comparison of **LLaMA2 7B**
> > > > >
> > > > > | Method                    | Acc. (%)         |
> > > > > |:---------------------------:|:--------------------:|
> > > > > |4-shot|$14.6$%|
> > > > > |MACM|$17.2$%|
> > > > >
> > > > > Table 4: Accuracy comparison of **LLaMA2 13B**
> > > > >
> > > > > | Method                    | Acc. (%)         |
> > > > > |:---------------------------:|:--------------------:|
> > > > > |4-shot|$28.7$%|
> > > > > |MACM|$32.1$%|
> > > > >
> > > > > Table 5: Accuracy comparison of **LLaMA3-Instruct 8B**
> > > > >
> > > > > | Method                    | Acc. (%)         |
> > > > > |:---------------------------:|:--------------------:|
> > > > > |4-shot|$30$%|
> > > > > |MACM|$33.7$%|
> > > > >
> > > > >
> > > > > **For *Weakness 3 (b)***:
> > > > >
> > > > > Thank you for your suggestion! In future versions, we will try to condense the text that repeats the content of **Figure 3** to make room for updates to the figures/tables that follow.
> > > > >
> > > > > **For *New Question***:
> > > > >
> > > > > Thank you for your suggestion. As mentioned in **For *Weakness 1 (3)***, we will clearly state this information in the paper: In our experiments, the GPT-4 model we used is `gpt-4-1106-preview`.
> > > > >
> > > > > As mentioned in our response to **reviewer upb5 in Question 3**, to ensure consistency in the testing version, we re-tested the performance of ToT on the 24-point game/sequence sorting using the `gpt-4-1106-preview` model from their official GitHub. For the MACM, the code interpreter is disabled. The results are shown in the table below:
> > > > >
> > > > > | Method                | Accuracy (%) |
> > > > > |:----------------------:|:------------:|
> > > > > |  IO               |      $23$    |
> > > > > |  CoT |      $25$       |
> > > > > | SC-CoT  |      $34$      |
> > > > > | ToT (b = 1)|     $55$       |
> > > > > | ToT (b = 5)|     $78$      |
> > > > > | MACM           |     $91$      |
> > > > >
> > > > > Experimental results show that MACM still outperforms ToT. By analyzing the error cases in ToT, we found that many problems that MACM can solve but ToT cannot are due to MACM's judge correcting these errors.
> > > > >
> > > > > ---
> > > > >
> > > > > Thank you again for your reply! And we welcome any further questions you may have.

---

> > > > > > ### Comment · Reviewer_pd1k · 2024-08-09
> > > > > >
> > > > > > Thanks for your reply! Below are my follow-up concerns:
> > > > > >
> > > > > > **For *Weakness 3 (a):*** The new format is better. Could you make similar revisions for other tables and figures for easier controlled variable comparison?
> > > > > >
> > > > > > **For *New Question:*** Could you provide exact versions of GPT-3.5 / GPT-4 / ... used? Please make everything comparable in a controlled variable way.
> > > > > >
> > > > > > **New weakness**:
> > > > > >
> > > > > > According to Figure 4, MACM's gains on open-source models are not clear and may be marginal. Could you compare MACM with 0-shot and Majority voting, at least on LLaMA3-Instruct 8B?

---

> > > > > > > ### Author Response · Authors · 2024-08-10
> > > > > > >
> > > > > > > Thank you for your reply and new suggestions!
> > > > > > >
> > > > > > > ---
> > > > > > >
> > > > > > > **For *Weakness 3 (a)***
> > > > > > >
> > > > > > > > Could you make similar revisions for other tables and figures for easier controlled variable comparison?
> > > > > > >
> > > > > > > Certainly, for **the 24-point game in Table 2**, we plan to break it down into several sub-tables, comparing based on prompting method/model/and whether code is used. The results are as follows:
> > > > > > >
> > > > > > > 1. Comparison of different Prompting Methods, ensure that the model is set to `gpt-4-1106-preview` and that the code interpreter is entirely disabled.
> > > > > > >
> > > > > > > | Method                | Accuracy (%) |
> > > > > > > |:----------------------:|:------------:|
> > > > > > > |  IO               |      $23$    |
> > > > > > > |  CoT |      $25$       |
> > > > > > > | SC-CoT (voter = $4$)  |      $34$      |
> > > > > > > | ToT (b = $1$)|     $55$       |
> > > > > > > | ToT (b = $5$)|     $78$      |
> > > > > > > | MACM           |     $91$      |
> > > > > > >
> > > > > > > 2. Test the impact of different models on MACM. Specifically, test MACM using `gpt-3.5-turbo-0125` and `gpt-4-1106-preview`, with the code interpreter disabled.
> > > > > > >
> > > > > > > | Model                | Accuracy (%) |
> > > > > > > |:----------------------:|:------------:|
> > > > > > > |  gpt-3.5-turbo-0125  |      $67$    |
> > > > > > > | gpt-4-1106-preview |      $91$     |
> > > > > > >
> > > > > > > 3. Assess the performance improvement of MACM when using the code interpreter, ensuring the model remains `gpt-4-1106-preview`.
> > > > > > >
> > > > > > > | Code interpreter usage | Accuracy (%) |
> > > > > > > |:----------------------:|:------------:|
> > > > > > > |  Yes  |      $99$    |
> > > > > > > | No |      $91$     |
> > > > > > >
> > > > > > > For **the Sequence sorting in Table 2** and **Table 3**, we will split it in the same way as the 24-point game in Table 2.
> > > > > > >
> > > > > > > **For *New Question***
> > > > > > >
> > > > > > > > Could you provide exact versions of GPT-3.5 / GPT-4 / ... used?
> > > > > > >
> > > > > > > Sure, all references to GPT-4 point to `gpt-4-1106-preview`, and all references to GPT-3.5 point to `gpt-3.5-turbo-0125`.
> > > > > > >
> > > > > > > **For *New weakness***
> > > > > > >
> > > > > > > > According to Figure 4, MACM's gains on open-source models are not clear and may be marginal.
> > > > > > >
> > > > > > > This is because open-source smaller models themselves lack sufficient intelligence, making them unable to handle structured prompts effectively. For example, in our analysis of some error cases with LLaMA3-Instruct 8B, we found that sometimes using few-shots not only fails to help solve the problem but also causes the model to repeatedly generate the prompts from the few-shots. This is particularly unfriendly for agent systems.
> > > > > > >
> > > > > > > > Could you compare MACM with 0-shot and Majority voting, at least on LLaMA3-Instruct 8B?
> > > > > > >
> > > > > > > Sure, the experimental details are as follows:
> > > > > > >
> > > > > > > ***Experiment Setup***
> > > > > > >
> > > > > > > We used the `vllm` library for model inference, enabling `flash-attention-2` to accelerate the attention computation process, with the `kv_cache` memory reserved to 95% of the total GPU memory. Inference was conducted using tensor parallelism. The experiments were run in parallel on 8 A100 GPUs (80 GB memory). The temperature was set to $0.1$, and the number of voters for majority voting was set to $4$.
> > > > > > >
> > > > > > > ***Experiment Results***
> > > > > > >
> > > > > > > | Method                    | Acc. (%)         |
> > > > > > > |:---------------------------:|:--------------------:|
> > > > > > > |0-shot|$22.7$%|
> > > > > > > |Majority Voting|$29.5$%|
> > > > > > > |MACM|$33.7$%|
> > > > > > >
> > > > > > > ***Experiment Conclusion***
> > > > > > >
> > > > > > > The experimental results show that compared to other prompting methods, MACM yields the most significant improvement for LLaMA3-Instruct 8B on the MATH dataset.
> > > > > > >
> > > > > > > ---
> > > > > > >
> > > > > > > Thank you again for your reply! And we welcome any further questions you may have.

---

> > > > > > > > ### Comment · Reviewer_pd1k · 2024-08-10
> > > > > > > >
> > > > > > > > Thanks for your reply! Below are my follow-up concerns:
> > > > > > > >
> > > > > > > > **For *New Question:*** You should also check the versions of GPTs used in baselines ... Please always try to be clearly comparable. It would be best if you could provide all related results at one time.
> > > > > > > >
> > > > > > > > **For *New Weakness:*** It is unreasonable to use a temperature as low as 0.1 for majority voting. Some more common choices are 0.3 or 0.7. It would be best if you could conduct a search. Besides, you should try to control the computation cost to be comparable (better in tokens). Given you have access to 8 A100 GPUs (80 GB memory), experiments on open-source models should be much cheaper and faster than ones on GPTs, which is a good way to resolve our concerns about the efficiency of the method.
> > > > > > > >
> > > > > > > > **New Weakness 2**: To test the impact of different models on **MACM as a new method**, it is not sufficient for GPT-3.5 to only test with MACM -- comparisons with baselines are needed. However, this could also be achieved with open-source models, which should be much cheaper and faster.

---

> > > > > > > > > ### Author Response · Authors · 2024-08-10
> > > > > > > > >
> > > > > > > > > Thank you for your reply!
> > > > > > > > >
> > > > > > > > > **For *New Question***
> > > > > > > > >
> > > > > > > > > As we mentioned in our previous response to you, we
> > > > > > > > >
> > > > > > > > > > ensure consistency in the testing version
> > > > > > > > >
> > > > > > > > > In all tests (including the baseline):
> > > > > > > > >
> > > > > > > > > > all references to GPT-4 point to gpt-4-1106-preview, and all references to GPT-3.5 point to gpt-3.5-turbo-0125
> > > > > > > > >
> > > > > > > > > **For *New Weakness***
> > > > > > > > >
> > > > > > > > > As per your request, we conducted new tests at temperatures of 0.3 and 0.7, and used the tokenizer of llama3-8B-Instruct to calculate the average number of input tokens and output tokens. The results are as follows:
> > > > > > > > >
> > > > > > > > > | Method                 |  Average Input Tokens | Average Output Tokens | Accuracy (%)|
> > > > > > > > > |:----------------------:|:------------:|:------------:|:------------:|
> > > > > > > > > |  0-shot  (voter = $32$)  |$65.8$	|	$513.1$	|	$22.7$	|
> > > > > > > > > |  Majority Voting  (t = $0.1$)  |$479.2$	|	$1179.2$	|	$29.5$	|
> > > > > > > > > |  Majority Voting  (t = $0.3$)  |$432.7$		|  $1019.2$	|	$31.2$	|
> > > > > > > > > |  Majority Voting  (t = $0.7$)  		| $526.1$		|	$1345.7$	|	$28.6$	|
> > > > > > > > > |    MACM   		|$784.7$	|	$1137.6$	|	$33.7$	|
> > > > > > > > >
> > > > > > > > > **For *New Weakness 2***
> > > > > > > > >
> > > > > > > > > Since prompting fundamentally relies on the model's inherent intelligence, our primary experiments, like previous prompting works such as ToT, CSV, and CR, are conducted on GPT-4. For GPT-3.5 and some open-source models, our supplementary experiments mainly provide validation for the extensibility of our method. Specifically for GPT-3.5, we did not merely test with MACM, but rather compared it with GoT in **Table 2**.

---

> ### Comment · Reviewer_pd1k · 2024-08-11
>
> Thanks for your reply! You've resolved almost all my concerns. I've updated my final comments in the official review.

---

### Official Review · Reviewer_zB7e · 2024-07-12

**Soundness:** 2
**Presentation:** 3
**Contribution:** 3
**Rating:** 5
**Confidence:** 3

**Summary:**

The paper presents a novel prompting technique MACM, which utilises multiple agents to cooperate and perform backtracking for mathematical reasoning problems.

**Strengths:**

The prompting method seems to work well for mathematical reasoning tasks and shows a degree of generalisation.

**Weaknesses:**

My main concern is with the evaluation method. As Appendix B suggest that the authors used GPT4 Turbo as a judge and only tested on a rather randomly selected set of MATH. This is very worrying, because (a) MATH has its own evaluation protocol, and the Minerva paper [1] also gave a good evaluation protocol. The reliance on GPT4 Turbo as a judge seems unjustifiable. (b) Why randomly select a subset, instead of using the whole MATH test set?


[1] Lewkowycz, A., Andreassen, A., Dohan, D., Dyer, E., Michalewski, H., Ramasesh, V., ... & Misra, V. (2022). Solving quantitative reasoning problems with language models. Advances in Neural Information Processing Systems, 35, 3843-3857.

**Questions:**

The MACM protocol seems rather convoluted to implement. Is there any venue to simplify it without losing performance in the authors' eyes?

**Limitations:**

The paper did not discuss much about the approach's limitation.

---

> ### Author Rebuttal · Authors · 2024-08-07
>
> Thank you for reviewing our work. Here are our responses to your concerns:
>
> ---
>
> **For weakness (a)**
>
> > MATH has its own evaluation protocol, and the Minerva paper [1] also gave a good evaluation protocol. The reliance on GPT4 Turbo as a judge seems unjustifiable.
>
> Thank you for your question! GPT-4 serves as a judge for voting/judging **intermediate steps** in the reasoning process, thereby helping the overall system achieve higher accuracy. It does **NOT** determine the correctness of the final result.
>
> In fact, our evaluation protocol follows the standards set in [1], Section D.1 of their paper. Initially, results are output in an easily extractable box and then verified using SymPy, as shown in **line 450 of our code in prompt/prompt.py**.
>
> **For weakness (b)**
>
> >  only tested on a rather randomly selected set of MATH. This is very worrying
>
> The MATH dataset contains tens of thousands of questions, and testing multiple methods on the entire dataset is often a costly endeavor. Even if testing one question with different prompting methods only costs 1 dollar, testing the entire dataset just once would require more than 12500 dollars, which is generally unaffordable. If a code interpreter is used, the cost would be even higher.
>
> We referred to the evaluation methods of other top-performing methods on the MATH dataset leaderboard and selected a subset of the data for testing. For example:
>
> 1. [2] includes the code implementation for the methods ranked **2nd** and **3rd** in accuracy on the leaderboard for the MATH dataset. Their results are based on a self-selected **1000 test cases**. **In their paper, Section 4.1 states: *We report results on the 1K-sized validation subsets for MATH and GSM8K created by us*.**
>
> 2. [3] includes the code implementation for the method ranked **4th** in accuracy on the leaderboard for the MATH dataset. Their results are based on a **500 test cases**. **In their paper, the first line of Table 5 states: * *denotes using 500 test examples subset*.**
>
> 3. In OpenAI's prompting-related paper [4], their evaluation results for the MATH dataset are also based on a self-selected **500 test cases**. **In their paper, Appendix C states: *We therefore evaluate our models only on the remaining 500 held-out problems. We selected these 500 test problems uniformly at random.***
>
> We randomly selected one-third of the data from the MATH dataset and ensured that the distribution of level 1-5 questions is consistent with the original dataset. To ensure a fair comparison, all self-implemented experiments were conducted on this one-third subset. The size of this subset already exceeds the 500 or 1000 cases chosen in previous work.
>
> **For Question**
>
> According to **Figure 7**, the four components of MACM each contribute to logical reasoning and solving complex problems. To adapt to problems of varying difficulty, we have set some external parameters, such as the number of voters and the number of Condition Mining iterations. For simpler problems that do not require complex logical reasoning, users can lower these parameters to effectively increase problem-solving speed.
>
> **Reference**
>
> [1] Solving quantitative reasoning problems with language models
>
> [2] OpenMathInstruct-1: A 1.8 Million Math Instruction Tuning Dataset
>
> [3] Cumulative Reasoning With Large Language Models
>
> [4] Let's Verify Step by Step
>
> ---
>
> Thank you again for your feedback! We hope our response can address your concerns.

---

> > ### Comment · Reviewer_zB7e · 2024-08-08
> >
> > Thank you for the clarifying comments.
> >
> > For weakness (a), I was looking at the sentence **We utilized the GPT-4 Turbo model (between January 1, 2024, and February 1, 2024) to test MACM’s performance on the MATH dataset.**
> > Does it mean that you are using it for inference not judging? In the Llama experiments, is GPT4 involved? At what capacity? This is very unclear to me.
> >
> > For weakness (b), I disagree with the premise that it takes 1 dollar to evaluate 1 question. For a vanilla zero-shot prompting method, back-of-the-envelope calculation: if a question takes 1000 input tokens and 500 output tokens, and using gpt4-turbo ($10/M input tokens, $30/M output tokens), the cost is 1000/1e6*10 + 500/1e6*30 = 0.025. So the total cost for the MATH test set (5000 datapoints) is 125 dollars. This makes it seem like at least for the final result, the whole MATH test set should be used. For other models, it's even cheaper.
> >
> > I understand MACM increases the inference-time tokens. But by how much? If it increases by a lot, shouldn't that be compared to majority voting in an iso-token manner?

---

> > > ### Author Response · Authors · 2024-08-09
> > >
> > > Thank you for your reply!
> > >
> > > ---
> > >
> > > **For *Weakness (a)***:
> > >
> > > Sorry for any confusion caused. There are a few points we want to clarify.
> > >
> > > 1. MACM is a multi-agent interaction framework introduced in our paper, where **each agent is played by an LLM**. The purpose of this framework is to **enhance the original LLM's ability in logical reasoning tasks**.
> > >
> > > 2. When we say: ***We utilized the GPT-4 Turbo model (between January 1, 2024, and February 1, 2024) to test MACM’s performance on the MATH dataset,*** it means that **we used GPT-4 Turbo to serve as the agent in our MACM framework**. We aim to compare our framework (MACM) with previous similar frameworks to demonstrate that our framework provides the most significant improvement to the original LLM's performance on logical reasoning tasks.
> > >
> > > 3. > Does it mean that you are using it for inference not judging?
> > >
> > >    Yes, we are conducting inference. However, our MACM framework includes a judging step. Please note that **this judging is intended to improve the overall system's performance** and will **NOT** be used as the final criterion for determining correctness. We need to compare the results with the ground truth to determine whether they are correct.
> > >
> > > 4. >  In the Llama experiments, is GPT4 involved?
> > >
> > >     No, In the LLaMA experiments, LLaMA will act as the agent within the MACM framework, and all decisions will be made by LLaMA.
> > >
> > > **For *Weakness (b)***:
> > >
> > > Thank you for your question. When you were evaluating cost, you used a zero-shot approach. Zero-shot means producing results without any prompts. Some common prompting methods include few-shots, Chain-of-Thought, Tree-of-Thought, Graph-of-Thought, etc.
> > >
> > > To give a simple example with few-shots, few-shots means including some similar questions as prompts before actually asking the model. **This will greatly increase the number of input tokens**, as each similar question is roughly the same length as the original question. A more complex example is Tree-of-Thought, which not only involves multi-step outputs but also requires few-shots prompting for each step. For more details, please refer to their official GitHub at `tree-of-thought-llm/src/tot/prompts/game24.py`.
> > >
> > > Although these methods require us to provide additional prompts during inference, they offer significant advantages, such as improving the accuracy of LLMs without the need for additional training, and enabling large models to output in specific formats, making it easier for us to extract specific information.
> > >
> > > For the statement:
> > >
> > > >  it takes 1 dollar to evaluate 1 question
> > >
> > > As indicated in **the second sentence of the first paragraph in *For weakness (b)* in our rebuttal**, this cost includes the expense of testing the same problem **using different prompting methods**. Since GPT-4 is continuously being updated, we need to retest previous prompting methods for a fair comparison. For example, in our **Table 4**, we need to remeasure each problem using IO, CoT, SC-CoT, and MACM.
> > >
> > > In fact, the cost is more than 1 dollar per question; testing about 4,200 cases cost us more than 6,000 dollars.
> > >
> > > > If it increases by a lot, shouldn't that be compared to majority voting in an iso-token manner?
> > >
> > > To test the trade-off between the accuracy and the responses generated by GPT-4 Turbo, we conducted a comparison in **Figure 6**. Additionally, in our response to **reviewer 4rbe's Weakness 1 & Question 1**, we compared the average number of responses between MACM and the baseline across different problems. The details are shown in the table below:
> > >
> > > **For MATH problem *(Our Table 1)***:
> > >
> > > | Method         | Average responses $n$|
> > > |:----------------:|:-----------------:|
> > > | CoT            | $5$               |
> > > | SC-CoT         | $25$              |
> > > | CSV†     | $7.552$                 |
> > > | CSV-Voting†     | $47.26$          |
> > > | MACM           | $40.48$           |
> > >
> > > †: They did not provide official code implementation and the related data is not involved in their paper. The data come from our own implementation.
> > >
> > > **For 24-point problem *(Our Table 2)***:
> > >
> > > | Method  | Average responses $n$ |
> > > |:---------:|:---------------------------:|
> > > |     CoT          |             $4.17$   |
> > > |     ToT* ($b=1$)  |          $23.72$     |
> > > | ToT* ($b=5$) | $67.81$                   |
> > > | MACM (w/o code verification)  | $63.3$  |
> > > | MACM (w code verification)    | $22.12$ |
> > >
> > > **For sequence sorting *(Our Table 2)***:
> > > | Method  | Average responses $n$|
> > > |:---------:|:---------------------------:|
> > > |     GoT*  |  $54.3$                  |
> > > | MACM (w/o code verification) | $51.72$|
> > > | MACM (w code verification)   | $6.882$ |
> > >
> > > *: The related data is not involved in their paper, we obtained the data from their official GitHub repository.
> > >
> > > ---
> > >
> > > Thank you again for your reply! And we welcome any further questions you may have.

---

> > > > ### Comment · Reviewer_zB7e · 2024-08-09
> > > >
> > > > Thank you for your reply. They have addressed my concerns well. I have bumped the score accordingly.
> > > >
> > > > Although, I do not think the question of "how much does MACM improve upon SC CoT if the tokens allowed are the same" is well answered. I've only seen the number of average responses and not the performances when token count is controlled. I would appreciate the effort to conduct such an experiment as I think this will significantly improve my confidence in the proposed method, but I understand if this is not feasible under the time/resource constraint.

---

> > > > > ### Comment · Reviewer_pd1k · 2024-08-09
> > > > >
> > > > > Thanks to the authors and the review zB7e for the efforts in the discussion.
> > > > >
> > > > > I agree that different inference-time methods should be compared under the same computation budget. A previous method might present weaker results only because it used a smaller budget. I will also consider this when rating.

---

> > > > > ### Author Response · Authors · 2024-08-10
> > > > >
> > > > > Thank you for your reply and reviewer pd1k's comments on this issue.
> > > > >
> > > > > ---
> > > > >
> > > > > To better address the question:
> > > > > > how much MACM improves upon SC-CoT when the allowed tokens are the same
> > > > >
> > > > > We conducted a detailed analysis on the 24 Game dataset, comparing SC CoT’s Input Tokens/Output Tokens and accuracy when the total cost is close to that of MACM. The detailed results are as follows:
> > > > >
> > > > > ***Experiment Setup***
> > > > >
> > > > > We tested SC-CoT with the number of voters set to $32$ and $40$. When the number of voters is $32$, the total cost of SC-CoT is less than that of MACM, while with $40$ voters, the total cost of SC-CoT exceeds that of MACM. The parameters used for MACM and the 24-point game dataset are consistent with those in **Table 2**. We used the model `gpt-4-1106-preview` and the code interpreter is disabled. The version of the `openai` library is `1.35.1`.The Input Tokens/Output Tokens/Total Cost/statistical code is as follows:
> > > > >
> > > > > ```
> > > > >         # Cost summary (Based on https://openai.com/api/pricing/)
> > > > >         input_tokens = completion.usage.prompt_tokens
> > > > >         output_tokens = completion.usage.completion_tokens
> > > > >         input_cost_per_million = 10.00   # $10.00 / 1M tokens
> > > > >         output_cost_per_million = 30.00  # $30.00 / 1M tokens
> > > > >         input_cost = (input_tokens / 1000000) * input_cost_per_million
> > > > >         output_cost = (output_tokens / 1000000) * output_cost_per_million
> > > > >         total_cost = input_cost + output_cost
> > > > >         accumulate_cost = total_cost + accumulate_cost
> > > > > ```
> > > > >
> > > > > ***Experiment Results***
> > > > >
> > > > > | Method                 | Total Cost (dollar) |  Total Input Tokens | Total Output Tokens | Accuracy (%)|
> > > > > |:----------------------:|:------------:|:------------:|:------------:|:------------:|
> > > > > |  SC-CoT  (voter = $32$)  |      $16.5636$  |	$897960$	|	$252800$	|	$43$	|
> > > > > |  SC-CoT  (voter = $40$)  |      $20.9034$|	$1126104$	|	$321411$	|	$47$	|
> > > > > |    MACM   		|	$20.5040$	|	$1697690$	|	$117569$	|	$91$	|
> > > > >
> > > > > ***Experiment Conclusion***
> > > > >
> > > > > Tests on the 24-point game indicate that when the costs are similar, MACM nearly doubles the accuracy compared to SC-CoT.
> > > > >
> > > > > ---
> > > > >
> > > > > Thank you again for your reply! And we welcome any further questions you may have.

---

> > > > > > ### Comment · Reviewer_pd1k · 2024-08-10
> > > > > >
> > > > > > Thanks for your reply!
> > > > > >
> > > > > > The experiment setting about cost is great! However, it is clear that ToT has great advantage over SC-CoT on 24-points game, which should be a better baseline. I recommend making similar comparisons with ToT. Besides, for SC-CoT, comparisons on MATH/SciBench/TheoremQA are more convincing for me. It is a pity that these results are not enough for me at the moment. Hopefully, you could always compare with the most competitive baselines and on the most meaningful tasks to your best.

---

> > > > > > > ### Author Response · Authors · 2024-08-10
> > > > > > >
> > > > > > > Thank you for your reply!
> > > > > > >
> > > > > > > For:
> > > > > > >
> > > > > > > > I recommend making similar comparisons with ToT.
> > > > > > >
> > > > > > > We have updated the table, and the ToT experiments were conducted according to their official GitHub. We used their default settings, except for updating the model to `gpt-4-1106-preview` to ensure consistency in testing. The result is as follow:
> > > > > > >
> > > > > > > | Method                 | Total Cost (dollar) |  Total Input Tokens | Total Output Tokens | Accuracy (%)|
> > > > > > > |:----------------------:|:------------:|:------------:|:------------:|:------------:|
> > > > > > > |  SC-CoT  (voter = $32$)  |      $16.5636$  |	$897960$	|	$252800$	|	$43$	|
> > > > > > > |  SC-CoT  (voter = $40$)  |      $20.9034$|	$1126104$	|	$321411$	|	$47$	|
> > > > > > > |  ToT  		|      $30.3382$	|	$1388479$	|	$548441$	|	$78$	|
> > > > > > > |    MACM   		|	$20.5040$	|	$1697690$	|	$117569$	|	$91$	|
> > > > > > >
> > > > > > > For:
> > > > > > >
> > > > > > > > Besides, for SC-CoT, comparisons on MATH/SciBench/TheoremQA are more convincing for me.
> > > > > > >
> > > > > > > We searched for all significant work related to prompting and found no previous detailed token-level statistics for the SC-CoT method on the entire MATH/SciBench/TheoremQA datasets. However, rerunning these three datasets would cost nearly ten thousand dollars. Therefore, we have calculated the cost of the SC-CoT method based on our experimental logs for **Figure 6**. The results are as follows:
> > > > > > >
> > > > > > > | Method                 | Total Cost (dollar) |  Total Input Tokens | Total Output Tokens | Accuracy (%)|
> > > > > > > |:----------------------:|:------------:|:------------:|:------------:|:------------:|
> > > > > > > |  SC-CoT  (voter = $20$)  |      $82.64$  |	$4869230$	|	$1131738$	|	$23.5$	|
> > > > > > > |    MACM   		|	$84.42$	|	$6932401$	|	$503276$	|	$84$	|

---

### Official Review · Reviewer_4rbe · 2024-07-12

**Soundness:** 3
**Presentation:** 3
**Contribution:** 3
**Rating:** 6
**Confidence:** 4

**Summary:**

This paper proposes a universal prompting method for solving complex reasoning problems such as mathematical problems and the 24-point game. The method first abstracts the conditions and objectives of the problems and then progressively discovers new conditions until enough information is gathered to solve the problem. Experiments on MATH, the 24-point game, SciBench, and TheoremQA indicate that the proposed method is effective and universal.

**Strengths:**

- The paper is well-structured and easy to follow.

- The experimental results are solid, showing significant improvements.

- The proposed method outperforms other prompting methods with a comparable number of responses.

**Weaknesses:**

- In Table 1 and Table 2, the number of responses for each problem should be highlighted to indicate whether the comparison is fair.

**Questions:**

- What is the number of responses for each model (including baselines and the proposed MACM) in Table 1 and Table 2?

- On which dataset is the experiment in Figure 6 conducted?

**Limitations:**

Yes

---

> ### Author Rebuttal · Authors · 2024-08-07
>
> Thank you for your valuable suggestions regarding our work. Here are our responses to the comments you raised:
>
> ---
>
> **For *Weaknesse 1 & Question 1***
>
> > What is the number of responses for each model (including baselines and the proposed MACM) in Table 1 and Table 2?
>
> Thanks for the question! In **Table 1**, for the methods we self-implemented, we standardized the number of responses $n$ generated by GPT-4 Turbo to $1$ for all prompting methods (as stated in **line 375** of our manuscript). For *SC-CoT*, we set the number of voters $v = 5$. The specific implementation method involves first asking the model to generate $v = 5$ different solution approaches for the problem, and then instructing the model to solve the problem according to each of these approaches. In Baseline *CSV + Voting*, their setting is: the number of sampled paths $k = 16$.
>
> Regarding the average number of responses $n$, we have summarized the results in the following table.
>
> | Method         | Average responses $n$|
> |:----------------:|:-----------------:|
> | CoT            | $5$               |
> | SC-CoT         | $25$              |
> | CSV†     | $7.552$                 |
> | CSV-Voting†     | $47.26$          |
> | MACM           | $40.48$           |
>
> †: They did not provide official code implementation and the related data is not involved in their paper. The data come from our own implementation.
>
> In **Table 2**, for the 24-point experiment, the data for *ToT* are sourced from their original paper, where the breadth b of the corresponding tree for *ToT* was set to $b=5$. In *SC-CoT*, the best of k samples parameter was set to $k=100$. For the sequence sorting experiment, the data for *GoT* also come from their original paper. They did not provide specific settings for this. For the *MACM*, we standardized the number of responses $n$ generated by GPT to $1$.
>
> Regarding the average number of responses $n$, we have summarized the results in the following table.
>
> **For 24-point problem**:
>
> | Method  | Average responses $n$ |
> |:---------:|:---------------------------:|
> |     CoT          |             $4.17$   |
> |     ToT* ($b=1$)  |          $23.72$     |
> | ToT* ($b=5$) | $67.81$                   |
> | MACM (w/o code verification)  | $63.3$  |
> | MACM (w code verification)    | $22.12$ |
>
> **For sequence sorting**:
> | Method  | Average responses $n$|
> |:---------:|:---------------------------:|
> |     GoT*  |  $54.3$                  |
> | MACM (w/o code verification) | $51.72$|
> | MACM (w code verification)   | $6.882$ |
>
> *: The related data is not involved in their paper, we obtained the data from their official GitHub repository.
>
> **For *Question 2***
>
> Thank you for your question! As stated in **lines 237-239**, we performed these two experiments on 200 randomly selected questions from the MATH dataset **that the original GPT-4 Turbo model answered incorrectly**. **Figure 6** demonstrates the corrective capabilities of different prompting methods under various parameter settings for these questions that GPT-4 originally answered incorrectly.
>
> ---
>
> Thank you again for all your feedback! We hope our answers can address all your concerns.

---

> > ### Comment · Reviewer_4rbe · 2024-08-13
> >
> > Thank you for your detailed responses, which have addressed most of my concerns. I believe I provided a fair rating and intend to maintain it.

---

### Decision · Program_Chairs · 2024-09-25

**Decision:**

Accept (poster)

**Comment:**

The paper presents a prompting technique, MACM, for mathematical reasoning problems. The reviewers found the paper easy to follow and the method working. An extensive discussion emerged, and all reviewers declared that most of their concerns were answered, with three out of four reviewers increasing their ratings. The lowest-scoring reviewer expressed a belief that `the submission has a contribution for proposing a new effective and general prompting method` and left his score as a reminder of presentation issues. It is for that reason I recommend the paper for acceptance.